# Rural and urban differences in the prevalence and determinants of Type-2 diabetes in Bangladesh

**Ashis Talukder**[1,2]\*, **Sabiha Shirin Sara**[1], **Md. Tanvir Hossain**[3], **Chuton Deb Nath**[4], **Riaz Rahman**[1], **Sadiq Hussain**[5], **Haribondhu Sarma**[6], **Md. Nazmul Huda**[7]

**1** Statistics Discipline, Science Engineering and Technology School, Khulna University, Khulna, Bangladesh, **2** National Centre for Epidemiology and Population Health, Australian National University, Canberra, Australia, **3** Sociology Discipline, Social Science School, Khulna University, Khulna, Bangladesh, **4** Mass Communication and Journalism Discipline, Social Science School, Khulna University, Khulna, Bangladesh, **5** Examination Branch, Dibrugarh University, Dibrugarh, India, **6** Research Fellow, National Centre for Epidemiology and Population Health, Australian National University, Canberra, Australia, **7** Translational Health Research Institute, Western Sydney University, Campbell Town, NSW, Australia

\* ashistalukder3168@ku.ac.bd, ashis.talukder@anu.edu.au

**Data Availability Statement:** Data are available in a public, open access repository. The BDHS 2017–2018 data are publicly accessible on request from

## Abstract

### Objective

To estimate the prevalence of Type 2 Diabetes (T2D) in urban and rural settings and identify the specific risk factors for each location.

### Method

We conducted this study using data from the 2017–18 Bangladesh Demographic and Health Survey (BDHS), sourced from the DHS website. The survey employed a stratified two-stage sampling method, which included 7,658 women and 7,048 men aged 18 and older who had their blood glucose levels measured. We utilized chi-square tests and ordinal logistic regression to analyze the association between various selected variables in both urban and rural settings and their relationship with diabetes and prediabetes.

### Results

The prevalence of T2D was 10.8% in urban areas and 7.4% in rural areas, while pre-diabetes affected 31.4% and 27% of the populations in these respective settings. The study found significant factors influencing diabetes in both urban and rural regions, particularly in the 55–64 age group (Urban: AOR = 1.88, 95% CI [1.46, 2.42]; Rural: AOR = 1.87, 95% CI [1.54, 2.27]). Highly educated individuals had lower odds of T2D, while wealthier and overweight participants had higher odds in both areas. In rural regions, T2D risk was higher among caffeinated drink consumers and those not engaged in occupation-related physical activity, while these factors did not show significant influence in urban areas. Furthermore, urban participants displayed a significant association between T2D and hypertension.

the DHS website at https://dhsprogram.com/data/available-datasets.cfm.

**Funding:** The authors received no specific funding for this work.

**Competing interests:** The authors have declared that no competing interests exist.

## Conclusion

Our study outlines a comprehensive strategy to combat the increasing prevalence of T2D in both urban and rural areas. It includes promoting healthier diets to control BMI level, encouraging regular physical activity, early detection through health check-ups, tailored awareness campaigns, improving healthcare access in rural regions, stress management in urban areas, community involvement, healthcare professional training, policy advocacy like sugary drink taxation, research, and monitoring interventions. These measures collectively address the T2D challenge while accommodating the distinct features of urban and rural settings.

## Introduction

Diabetes mellitus or type 2 diabetes (T2D), a chronic and systemic condition, significantly contributes to a spectrum of severe health outcomes, including cardiovascular diseases, strokes, vision impairments, neuropathies, kidney disorders, and the necessity for limb amputations. The incidence of T2D is increasing at a fast pace on a worldwide scale, with a special focus on countries such as Asia, the Middle East, and North Africa [1]. Projections spanning 2010 to 2030 anticipate a significant 69% rise in the prevalence of diabetes in developing countries, whereas developed countries are expected to have a comparatively lower increase of 20% [2]. By 2045, it is predicted that 700 million individuals worldwide will have diabetes, marking a 51% increase from 2019 [3]. Concurrently, the incidence of pre-diabetes among adults is projected to rise from 374 million (7.5% of the population) to 548 million (8.6% of the population) by 2045 [3]. Furthermore, individuals with T2DM experience an average reduction in life expectancy of approximately 10 years, with 80% of them succumbing to cardiovascular complications, indicating the significant impact of T2DM on mortality and morbidity [4]. T2D is also listed as the tenth leading factor influencing life expectancy [5]. Additionally, T2D heightens the risk of conditions of substantial clinical importance, including dementia [6], a twofold increase in cancer risk [7,8], and an elevated propensity for cardiovascular diseases [9]. Psychological ailments like depression and physiological disturbances such as platelet dysfunction are also associated with T2DM [10,11].

Due to dietary choices and lifestyle factors, T2D is becoming increasingly prevalent in South Asia [12]. Bangladesh, known for its high population density, ranks second in the region with a 6.31% prevalence of diabetic adults [13]. In 2019, 8.4 million people in Bangladesh had diabetes, and this number is projected to double to 15.0 million by 2045 [14]. Additionally, it is estimated that 3.8 million individuals had pre-diabetes in Bangladesh in 2019 [14]. Between 2011 and 2018, the prevalence of diabetes increased among adults aged 35 and older, rising from 10.95% to 13.75% [15]. Pre-diabetes is also on the rise, as indicated by numerous studies in Bangladesh, and is accompanied by low rates of treatment and control [13,16]. Unfavorable living conditions, especially crowded urban slums and associated stress, may contribute to the growing incidence of T2D [17]. Consequently, investigating the differences in T2D prevalence and related risk factors between urban and rural populations is an intriguing area of study.

In Bangladesh, there is a growing body of studies examining T2D and its risk factors. Several studies have highlighted factors such as education level, hypertension, financial status, physical activity, abdominal obesity, social class, family history, waist-hip ratio, and urbanization as contributors to diabetes in Bangladesh [18–21]. Additionally, gender [6,8,9], high blood pressure [15], older age [22], lifestyle choices [11,23], BMI [24], and ethnicity [25] have

also been identified as diabetes risk factors in various studies. It is worth noting that there exists evidence indicating an annual increase in diabetes prevalence in both urban and rural settings [26]. However, there exists a dearth of comprehensive studies aimed at estimating diabetes prevalence and its concomitant risk determinants via nationally representative surveys, particularly in segregating urban and rural domains. Hence, our study aims to estimate the prevalence of T2D in both urban and rural areas and identify the specific risk factors relevant to each location.

## Methods

### Data source

In this study, we utilized secondary data extracted from the Bangladesh Demographic and Health Survey (BDHS) conducted during the 2017–18 period. This extensive dataset, meticulously administered by the National Institute for Population Research and Training (NIPORT), is openly accessible on the DHS program website [27]. For analysis, we employed the existing variable "Type of place of residence" to partition the dataset into discrete urban and rural categories. This categorization serves as the foundation for our research, enabling us to conduct a focused investigation into the prevalence of type 2 diabetes and its associated risk factors within the distinct urban and rural settings of Bangladesh.

### Sampling design

The BDHS 2017–18 survey employed a stratified two-stage random sampling approach. Initially, the Bangladesh Bureau of Statistics (BBS) conducted the sample selection using probability proportionate to size, considering geographical areas as the basis. Subsequently, a comprehensive household census was conducted in all selected enumeration areas (EAs) during the second stage to establish a structured sampling framework. The 2017–18 sample comprised 20,250 households, and interviews were successfully completed by 19,457 of them. Following the exclusion of three clusters affected by flood-induced erosion, the study concluded with a total of 672 clusters [27]. In this study, it was observed that among the eligible participants, 87% of women and 80% of men aged 18 years or older had their blood glucose levels measured. These participants amounted to 7,658 women and 7,048 men [27].

### Selection of sample

Fig 1 provides a visual representation of our sample selection process. From a total participant pool of 20,127 individuals, a subgroup of 12,300 met the eligibility criteria for diabetes measurement. These eligible participants were further stratified into two distinct categories: Urban (comprising 4,393 individuals) and Rural (encompassing 7,907 individuals), based on their respective places of residence. This categorization served as the foundation for our subsequent analytical investigations.

### Dependent variable

The responders were instructed to fast for at least eight hours before to the test in order to test their fasting plasma blood glucose levels. The acquired value was converted to fasting plasma glucose equivalent values using the HemoCue Glucose 201 Dm system [28]. Then, in our study, we classified the fasting plasma glucose values into three categories in accordance with the recommendations of the World Health Organisation (WHO), and we called the dependent variable "Diabetes Status" in order to determine the presence of diabetes. Here, fasting blood glucose levels between 70 mg/dL (3.9 mmol/L) and 100 mg/dL (5.6 mmol/L) were regarded as

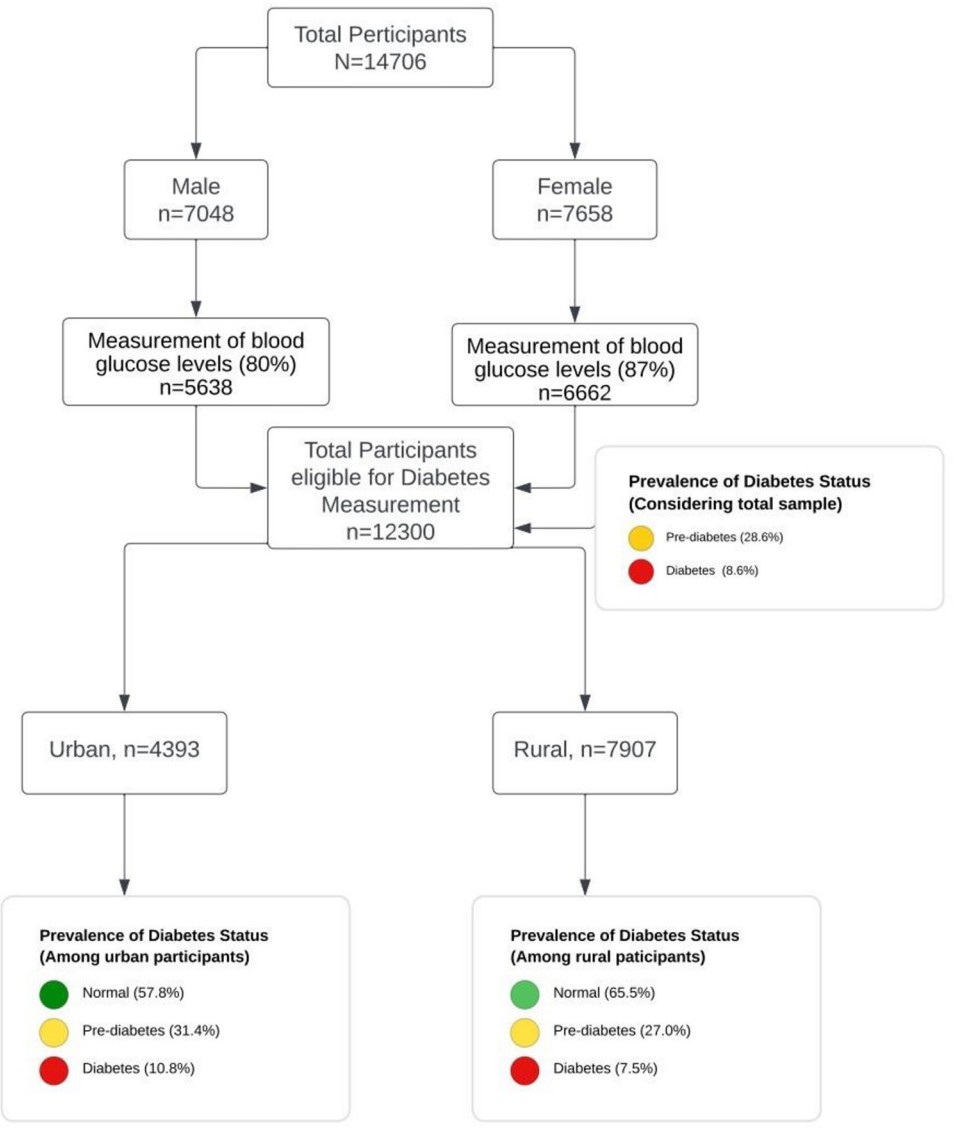

**Fig 1. Flow chart showing sample selection procedure.**

normal, while those between 100 and 125 mg/dL (5.6 to 6.9 mmol/L) were classified as prediabetes and those above 126 mg/dL (7 mmol/L) as diabetic following the guidelines of Americans Diabetes Association (ADA) [19,29].

## Explanatory variables

Gender, age, education level, wealth index, smoking habit, consumption of caffeinated beverages, physical activity, hypertension, and BMI were explanatory variables. The selection of these factors and their categorisation was made after a thorough assessment of earlier studies from Bangladesh and other countries. Within the framework of the Bangladeshi education system, "Primary education" denotes the successful completion of grades 1 to 5, "Secondary education" includes grades 6 to 12, and progression beyond grade 12 is designated as "Higher Education." Age is a numerical variable that has been broken down into the following four

groups: youth (18–24 years) [30], prime working age (25–54 years), mature working age (55–64 years), and elder (65 years and above) [31]. We created the categorical variable "occupational physical activity status" based on the job description of the respondents [32,33]. If a respondent's occupation involved physical labor or non-sedentary work, we classified them as individuals engaged in physical activity. This category encompasses various occupations, including farmers, fishermen, rickshaw drivers, poultry and livestock raisers, bricklayers, construction workers, road builders, boatmen, and factory workers [32]. Another variable BMI was calculated by the DHS program. They collected data on heights and weights. Heights were measured standing up and weights were measured using SECA scales with a digital display. Then, BMI was calculated as weight in kilograms divided by the square of height in meters. To make our study more meaningful, we then categorized BMI into three categories. In this case, the Asia-Pacific BMI classification had been followed to classify more accurately. BMI with 18.5 kg/m$^2$ was considered underweight, BMI from 18.5 to 22.9 was normal, and BMI $\geq 23$ was overweight or obese [34]. Again, the measurement of blood pressure was taken at three different times, at about 10 min intervals, by trained health technicians. Then, the average of the measurements was granted as the final measurement for BP. The respondents with systolic blood pressure (SBP) < 140 and diastolic blood pressure (DBP) < 90 mmHg were considered to have no hypertension. If one violates the above rules, he/she has hypertension [35].

## Statistical analysis

We have performed both bivariate and multivariable analysis. Chi-square had been used to find out the bivariate association between diabetes status and other explanatory variables and ordinal regression analysis had been used as the multivariable analysis.

Let, Y be the ordinal dependent variable with j categories (j>2) and $x_1, x_2, \ldots, x_p$ be the independent variables. Moreover, $\beta_j$ (for j = 1, 2,.., j-1) be the coefficients associated with the independent variables for the j-th cumulative category. The cumulative probabilities $P(Y \leq j)$ for each category j are modeled as follows:

$$P(Y \leq j) = P(Y = 0) \; if \; j = 0$$

$$P(Y \leq j) = P(Y = 1) + P(Y = 0) \; if \; j = 1$$

$$P(Y \leq j) = P(Y = 2) + P(Y = 1) + P(Y = 0) \; if \; j = 2$$

$$P(Y \leq j) = 1 \; if \; j = j - 1$$

Now, the cumulative log-odds (logit) of the probabilities:

$$\eta_j = ln \left[ \frac{P(Y \leq j)}{P(Y > j)} \right] = \beta_{0j} + \beta_1 X_1 + \beta_2 X_2 + \cdots + \beta_p X_p$$

## Statistical tools

The study requires model adjustment to detect risk factors for pre-diabetes and diabetes and raise awareness about lifestyle. In univariate analysis, a frequency distribution table was used to describe data. Chi-square test was used to find the association between variables. In multivariable analysis, an ordinal logistic regression model was fitted. The data analysis was conducted using R (version 4.1.0). A 95% confidence interval was used to interpret the regression analysis with a significance level set at $p<0.05$.

## Results

### Background characteristic of study participants

Table 1 provides an overview of the background characteristics of participants in both urban and rural settings. In urban areas, male participants constituted a higher proportion (53.2%), while rural areas had a greater representation of female participants (55.3%). Educational attainment differed significantly, with 24.7% of urban participants classified as highly educated compared to a lower prevalence of 12% among their rural counterparts. When it came to the consumption of caffeinated beverages, urban residents exhibited a higher percentage (9.6%) compared to their rural counterparts (6.4%), whereas smoking was more prevalent among rural individuals (17%) as opposed to urban dwellers (13.4%). Notably, hypertension (25.1%) and overweight (50.2%) were more prevalent among urban participants, whereas rural partici-pants (43%) reported a higher rate of physical inactivity. In terms of wealth distribution, 43.7% of urban participants were categorized as the richest, whereas in rural areas, only 9.8% fell into

**Table 1. Characteristics of the study participants.**

| Covariates | Category | Urban | | Rural | |
|---|---|---|---|---|---|
| | | Frequency | Percentages | Frequency | Percentages |
| **Gender** | Male | 2934 | 53.2 | 4114 | 44.7 |
| | Female | 2577 | 46.8 | 5081 | 55.3 |
| **Age** | Youth | 1176 | 21.3 | 1828 | 19.9 |
| | Prime working | 3327 | 60.4 | 5228 | 56.9 |
| | Mature working | 556 | 10.1 | 1105 | 12.0 |
| | Older | 452 | 8.2 | 1034 | 11.2 |
| **Education level** | No education | 1043 | 18.9 | 2668 | 29.0 |
| | Primary | 1467 | 26.6 | 2861 | 31.1 |
| | Secondary | 1636 | 29.7 | 2557 | 27.8 |
| | Higher | 1361 | 24.7 | 1099 | 12.0 |
| **Caffeinated drink** | No | 4317 | 90.4 | 7812 | 93.5 |
| | Yes | 461 | 9.6 | 541 | 6.5 |
| **Smoking Status** | No | 4137 | 86.6 | 6931 | 83.0 |
| | Yes | 641 | 13.4 | 1421 | 17.0 |
| **Division** | Barisal | 2966 | 9.2 | 6582 | 11.4 |
| | Chittagong | 4923 | 15.2 | 8413 | 14.6 |
| | Dhaka | 7409 | 22.9 | 5387 | 9.4 |
| | Khulna | 3949 | 12.2 | 7016 | 12.2 |
| | Mymensingh | 2541 | 7.9 | 7564 | 13.1 |
| | Rajshahi | 3497 | 10.8 | 7276 | 12.6 |
| | Rangpur | 3189 | 9.9 | 7526 | 13.1 |
| | Sylhet | 3817 | 11.8 | 7764 | 13.5 |
| **Hypertension** | No | 3577 | 74.9 | 6331 | 75.8 |
| | Yes | 1199 | 25.1 | 2024 | 24.2 |
| **BMI** | Underweight | 599 | 12.6 | 1609 | 19.5 |
| | Normal | 1762 | 37.1 | 3658 | 44.3 |
| | Overweight | 2382 | 50.2 | 2990 | 36.2 |
| **Occupational physical activity status** | Yes | 1359 | 75.1 | 3919 | 57.0 |
| | No | 4089 | 24.9 | 5203 | 43.0 |
| **Wealth status** | Poorest | 2951 | 9.1 | 15110 | 26.3 |
| | Poorer | 2698 | 8.4 | 14601 | 25.4 |
| | Middle | 4359 | 13.5 | 12688 | 22.1 |
| | Richer | 8187 | 25.4 | 9517 | 16.5 |
| | Richest | 14096 | 43.7 | 5612 | 9.8 |
| **Diabetes Status** | Normal | 2539 | 57.8 | 5182 | 65.5 |
| | Pre-diabetes | 1381 | 31.4 | 2138 | 27.0 |
| | Diabetes | 473 | 10.8 | 587 | 7.4 |

this category. Diabetes exhibited a higher prevalence among urban participants (10.8%) in contrast to rural participants (7.4%).

## Assessing association between T2D and selected covariates

Table 2 presents a comprehensive analysis of the relationship between socio-demographic variables and T2D. Age exhibited a significant association with T2D in both urban ($\chi^2$ = 93.232, $p<0.01$) and rural ($\chi^2$ = 93.232, $p<0.01$) contexts, with working-age individuals displaying the

**Table 2. Bivariate analysis of the selected variables for urban and rural areas.**

| Covariates | Urban | | | | Rural | | | |
|---|---|---|---|---|---|---|---|---|
| | Blood Glucose | | | | Blood glucose | | | |
| | Normal | Pre-diabetes n (%) | Diabetes n (%) | Chi-square (p-value) | Normal | Pre-diabetes n (%) | Diabetes n (%) | Chi-square (p-value) |
| **Gender** | | | | | | | | |
| Male | 1152(45) | 597(43) | 200(42) | 2.5953 (0.2732) | 2198(42) | 886(41) | 265(45) | 2.6134 (0.2707) |
| Female | 1387(55) | 784(57) | 273(58) | | 2983(58) | 1252(59) | 322(55) | |
| **Age** | | | | | | | | |
| Young | 610(24) | 272(20) | 37(8) | **89.398 (<0.001)** | 1129(22) | 356(17) | 50(9) | **93.232 (<0.001)** |
| Working | 1523(60) | 847(61) | 300(63) | | 2910(56) | 1279(60) | 344(59) | |
| Primary working | 210(8) | 146(11) | 74(16) | | 582(11) | 273(13) | 109(19) | |
| Older | 196(8) | 116(8) | 62(13) | | 560(11) | 230(11) | 84(14) | |
| **Education level** | | | 87(18) | 5.0437 (0.5382) | | | | 3.8792 (0.693) |
| No education | 498(20) | 268(19) | 124(26) | | 1513(29) | 599(28) | 165(28) | |
| Primary | 697(27) | 390(28) | 159(34) | | 1619(31) | 678(32) | 197(34) | |
| Secondary | 782(31) | 396(29) | 103(22) | | 1430(28) | 610(29) | 166(28) | |
| Higher | 562(22) | 327(24) | | | 619(12) | 251(12) | 59(10) | |
| **Caffeinated drink** | | | | **7.9886 (0.018)** | | | | **13.478 (0.001)** |
| No | 2323(92) | 1258(91) | 412(87) | | 4893(95) | 1974(92) | 545(93) | |
| Yes | 215(8) | 123(9) | 59(13) | | 282(5) | 163(8) | 42(7) | |
| **Smoking status** | | | | 1.5009 (0.4721) | | | | 1.6516 (0.4379) |
| Yes | 2207(87) | 1215(88) | 405(86) | | 4326(84) | 1788(84) | 478(82) | |
| No | 331(13) | 166(12) | 66(14) | | 849(16) | 349(16) | 108(18) | |
| **Division** | | | | **270.2 (<0.01)** | | | | **110.8 (<0.01)** |
| Barisal | 237(9) | 140(10) | 46(10) | | 537(10) | 256(12) | 65(11) | |
| Chittagong | 368(14) | 198(14) | 66(14) | | 633(12) | 302(14) | 100(17) | |
| Dhaka | 298(12) | 425(31) | 142(30) | | 400(8) | 256(12) | 83(14) | |
| Khulna | 394(16) | 163(12) | 65(14) | | 743(14) | 271(13) | 70(12) | |
| Mymensingh | 243(10) | 96(7) | 30(6) | | 691(13) | 271(13) | 66(11) | |
| Rajshahi | 369(15) | 123(9) | 46(10) | | 745(14) | 255(12) | 75(13) | |
| Rangpur | 327(13) | 117(8) | 38(8) | | 808(16) | 245(11) | 47(8) | |
| Sylhet | 303(12) | 119(9) | 40(8) | | 624(12) | 282(13) | 81(14) | |
| **Hypertension** | | | | **80.643 (<0.01)** | | | | **76.271 (<0.01)** |
| No | 1967(78) | 1034(75) | 273(58) | | 3984(77) | 1640(77) | 357(61) | |
| Yes | 569(22) | 347(25) | 198(42) | | 1193(23) | 498(23) | 230(39) | |
| **BMI** | | | | **87.431 (<0.01)** | | | | **91.688 (<0.01)** |
| Underweight | 376(15) | 147(11) | 35(8) | | 1070(21) | 391(19) | 67(12) | |
| Normal | 1024(41) | 486(35) | 126(27) | | 2323(45) | 898(43) | 206(36) | |
| Overweight | 1114(44) | 740(54) | 302(65) | | 1726(34) | 822(39) | 303(53) | |
| **Occupational physical activity status** | | | | | | | | |
| **Yes** | 750(30) | 350(25) | 82(17) | **32.896 (<0.01)** | 2400(47) | 883(41) | 193(33) | **46.357 (<0.01)** |
| **No** | 1776(70) | 1025(75) | 389(83) | | 2761(53) | 1247(59) | 389(67) | |
| **Wealth status** | | | | **227.21 (<0.01)** | | | | **184.27 (<0.01)** |
| Poorest | 300(12) | 64(5) | 18(4) | | 1396(27) | 514(24) | 99(17) | |
| Poorer | 276(11) | 84(6) | 17(4) | | 1365(26) | 488(23) | 106(18) | |
| Middle | 444(17) | 176(13) | 48(10) | | 1189(23) | 457(21) | 121(21) | |
| Richer | 654(26) | 357(26) | 109(23) | | 790(15) | 382(18) | 125(21) | |
| Richest | 865(34) | 700(51) | 473(59) | | 441(9) | 297(14) | 136(23) | |

highest T2D percentages compared to other age groups in both settings. Caffeinated drink consumption also showed a noteworthy association with diabetes in both urban ($\chi^2$ = 7.988, $p < 0.05$) and rural ($\chi^2$ = 13.478, $p < 0.01$) areas, with lower diabetes prevalence observed among participants who consumed caffeinated beverages, at 13% in urban and 7% in rural areas, respectively. The division, encompassing geographical areas, was significantly linked to diabetes in both urban ($\chi^2$ = 270.2, $p < 0.01$) and rural ($\chi^2$ = 110.80, $p < 0.01$) regions. Specifically, Dhaka exhibited the highest diabetes (30%) and pre-diabetes (31%) rates in urban areas, whereas Chittagong recorded the highest diabetes (17%) and pre-diabetes (14%) rates in rural areas. Hypertension emerged as another significant risk factor associated with diabetes, demonstrating statistical significance in both urban ($\chi^2$ = 80.643, $p < 0.01$) and rural ($\chi^2$ = 76.271, $p < 0.01$) environments. The prevalence of hypertension among T2D participants was higher in urban (42%) than rural (39%) settings. Furthermore, body mass index (BMI), physical activity, and wealth status displayed significant associations with diabetes status in both urban and rural settings. Overweight prevalence was relatively higher among urban T2D participants (65%) compared to rural counterparts (53%). Additionally, 83% of urban T2D participants reported physical inactivity, while the prevalence was lower among rural participants (67%). Concerning wealth status, diabetes prevalence was higher among the richest urban participants (59%) compared to the wealthiest rural participants (23%). Conversely, pre-diabetes prevalence was higher among the poorest rural participants (24%) than their urban counterparts (5%).

### Determinants of T2D in Urban and rural settings

Table 3 presents the estimates of ordinal logistic regression analysis, aimed at identifying the risk factors associated with T2D in both urban and rural settings. The analysis revealed that the odds of developing T2D were significantly higher among individuals in the primary working age group (AOR = 1.35, 95% CI [1.14, 1.59]), the mature working age group (AOR = 1.88, 95% CI [1.46, 2.42]), and the older age group (AOR = 1.64, 95% CI [1.25, 2.16]) when compared to youth participants in urban regions. Similar findings were observed in rural areas. Furthermore, higher education was linked to reduced T2D risk in both urban (AOR = 0.67, 95% CI [0.54,0.84]) and rural (AOR = 0.80, 95% CI [0.66, 0.97]) regions. In rural regions, individuals who consumed caffeinated drinks (AOR = 1.24, 95% CI [1.02, 1.50]) and those who were not engaged in occupation-related physical activity (AOR = 1.29, 95% CI [1.16, 1.43]) had a higher odds of developing T2D. In urban regions, middle-income participants had 1.76 times higher odds, while richer (AOR = 2.64, 95% CI [2.01, 3.52]) and richest (AOR = 4.16, 95% CI [3.15, 5.54]) individuals had even greater odds of T2D. In rural settings, the odds were 1.39 times higher among richer participants and 2.15 times higher among the richest ones compared to the poorest. Furthermore, being overweight was associated with higher odds of T2D in both urban (AOR = 1.39, 95% CI [1.13, 1.72]) and rural (AOR = 1.31, 95% CI [1.14, 1.52]) regions. Finally, urban participants with hypertension had increased odds (AOR = 1.25, 95% CI [1.08, 1.45]) of developing T2D, while there was no significant effect among rural hypertensive participants.

### Discussion

Our findings illustrate substantial disparities in the prevalence of T2D and prediabetes between urban and rural areas, with a focus on sociodemographic risk factors such as age, education level, wealth status, physical activity, consumption of caffeinated beverages, hypertension, and BMI. Specifically, the prevalence of T2D and prediabetes in urban settings is 10.8% and 31.4%, respectively, whereas in rural areas, it is 7.4% and 27%, respectively, underscoring

**Table 3. Predictors of T2DM across urban and rural areas.**

| Variables | Urban | | Rural | |
|---|---|---|---|---|
| | AOR (95%CI) | p-value | AOR | p-value |
| Intercept ($\alpha_1$) | 1.21 (1.15, 1.62) | <0.001 | 0.85 (0.51, 0.82) | <0.001 |
| Intercept ($\alpha_2$) | 1.32 (1.21, 1.81) | <0.001 | 0.91 (0.66, 0.94) | <0.001 |
| **Gender** | | | | |
| Male (ref) | - | - | | |
| Female | 1.06 (0.93, 1.20) | 0.344 | 0.95 (0.86,1.05) | 0.408 |
| **Age** | | | | |
| Youth (ref) | - | - | | |
| Prime working | 1.35 (1.14, 1.59) | <0.001 | 1.56 (1.35,1.79) | <0.001 |
| Mature working | 1.88 (1.46, 2.42) | <0.001 | 1.87 (1.54, 2.27) | <0.001 |
| Older | 1.64 (1.25, 2.16) | <0.001 | 1.52 (1.23, 1.87) | <0.001 |
| **Education level** | | | | |
| No education (ref) | - | - | | |
| Primary | 1.01 (0.83,1.21) | 0.966 | 1.09 (0.96, 1.24) | 0.139 |
| Secondary | 0.82 (0.68,1.01) | 0.056 | 0.99 (0.86, 1.14) | 0.911 |
| Higher | 0.67 (0.54,0.84) | <0.001 | 0.80 (0.66, 0.97) | 0.028 |
| **Caffeinated drink** | - | - | | |
| No (ref) | | | | |
| Yes | 1.09 (0.88,1.35) | 0.416 | 1.24 (1.02, 1.50) | 0.025 |
| **Smoking status** | | | | |
| No (ref) | - | - | | |
| Yes | 0.97 (0.81,1.18) | 0.810 | 0.98 (0.86, 1.12) | 0.827 |
| **BMI** | | | | |
| Underweight (ref) | - | - | | |
| Normal | 1.09 (0.88,1.34) | 0.400 | 1.09 (0.96, 1.25) | 0.166 |
| Overweight | 1.39 (1.13,1.72) | 0.002 | 1.31 (1.14, 1.52) | <0.001 |
| **Wealth status** | | | | |
| Poorest (ref) | - | - | | |
| Poorer | 1.25 (0.89,1.76) | 0.186 | 0.97 (0.84, 1.11) | 0.698 |
| Middle | 1.76 (1.31,2.38) | <0.001 | 1.07 (0.93,1.23) | 0.336 |
| Richer | 2.64 (2.01,3.52) | <0.001 | 1.39 (1.19, 1.63) | <0.001 |
| Richest | 4.16 (3.15,5.54) | <0.001 | 2.15 (1.80, 2.57) | <0.001 |
| **Occupational physical activity status** | | | | |
| Yes (ref) | - | - | | |
| No | 1.09 (0.93,1.26) | 0.253 | 1.29 (1.16, 1.43) | <0.001 |
| **Hypertension** | | | | |
| No (ref) | - | - | | |
| Yes | 1.25 (1.08,1.45) | 0.002 | 1.09 (0.97, 1.22) | 0.125 |

Note. Estimates obtained from ordinal logistic regression; AOR: Adjusted odds ratio; BMI: Body mass index; 95% CI: 95%Confidence Interval.

the lower prevalence of diabetes and prediabetes in rural settings compared to urban areas. Moreover, our study reveals that individuals residing in urban environments face twice the risk of developing diabetes when compared to their rural counterparts.

Our study found that older people, especially those in their working-age and older age, have a higher chance of getting diabetes compared to younger individuals in both urban and rural areas in Bangladesh. This might be because as people age, their bodies might become less able

to process sugar properly, and they might not be as physically active as they used to be [36,37]. The fact that age is linked to diabetes in both urban and rural areas shows that it is an important factor no matter where people live or what their lifestyle is like. This means we need to focus on helping older people prevent and manage diabetes with targeted interventions. Our findings also highlight the importance of healthcare systems preparing for more cases of diabetes in older people by providing the necessary resources and tailored care to address the health challenges associated with diabetes in this age group.

Our study findings demonstrate an inverse relationship between higher education levels and the incidence of diabetes, both in urban and rural contexts among Bangladeshi individuals. This observation aligns with prior research [38] and implies that education may serve as a protective factor against diabetes by fostering greater health awareness and promoting healthier lifestyle choices [39]. These results underscore the pivotal role of education in the prevention and management of diabetes, emphasizing the importance of educational programs aimed at enhancing health literacy among the Bangladeshi population. Given the apparent significance of higher education in reducing the risk of diabetes, it is essential for policies and interventions to consider educational attainment as a key component in addressing the escalating diabetes burden in Bangladesh and in fostering healthier behaviors across all demographic groups [40]. It is important to note that while education may not directly reduce diabetes, it can substantially improve health literacy and enhance awareness of diabetes-related complications and adherence to dietary recommendations [40,41]. Further research is warranted to gain deeper knowledge about the mechanisms underlying this relationship and to tailor educational interventions effectively for diverse urban and rural communities.

Our study reveals that individuals classified as overweight faced a substantially higher likelihood of developing diabetes, irrespective of their urban or rural residence. This aligns with what other studies have found [39], which show that a person's body weight, measured by BMI, can strongly predict the chances of T2D. A study conducted in Bangladesh also supports this finding [42]. One reason for this connection could be the dietary behaviour–consuming high-calorie and low-nutrition foods, which can lead to insulin-induced weight gain that substantially suppresses blood sugar control [43]. Additionally, being inactive, which means not getting enough physical activity, is common in both urban and rural areas and can make the risk of diabetes even higher for people who are overweight [44,45]. Public health programs for maintaining a healthy weight, and making healthier food and exercise choices urgently warrant and should be adapted to the different needs of people living in urban and those in rural areas in Bangladesh.

Our study identified a significant association between higher income status and an elevated risk of T2D in both urban and rural settings. Specifically, individuals classified as richer, the richest, or with middle income residing in urban areas faced a significantly increased likelihood of developing T2D. These findings underscore the complex interplay between socioeconomic factors and diabetes risk, transcending geographical boundaries. While urban environments often offer improved access to healthcare services and education, they also frequently entail lifestyle changes, including dietary shifts and reduced physical activity, which can contribute to higher diabetes risk among individuals with greater economic resources [46,47]. Moreover, in urban settings, the concentration of economic opportunities and resources might expose middle-income individuals to similar dietary and lifestyle patterns as their wealthier counterparts [48]. These results emphasize the need for targeted public health interventions that consider income-related disparities in diabetes risk, particularly in urban contexts. Effective strategies for diabetes prevention and management should address the multifaceted socioeconomic determinants of this chronic condition, aiming to reduce its burden across diverse income groups in both urban and rural areas.

In rural areas, we found a strong connection between T2D and people who drink caffeinated beverages, are less physically active, and individuals with hypertension in urban regions. Caffeinated drinks might contribute to diabetes risk through their potential to affect insulin sensitivity or promote unhealthy dietary habits [49]. Lower physical activity levels in rural areas may result from different occupational and lifestyle factors, potentially leading to weight gain and insulin resistance [45,50]. On the other hand, in urban areas, we noticed a significant link between high blood pressure and diabetes. This could be because of the stress, eating habits, and lack of physical activity often seen in city life [45,51,52]. These findings highlight the necessities of different strategies for urban and rural individuals. For rural populations, it is crucial to promote reduced consumption of caffeinated drinks and encourage increased engagement in physically active occupations. Meanwhile, in urban areas, a key focus should be on controlling blood pressure.

This study also identifies that prediabetes affects a significant portion of the Bangladeshi population in both urban and rural areas. Managing prediabetes can help control the higher prevalence of T2D [53]. Additionally, there are several important advantages to controlling prediabetes in Bangladesh for both individuals and the healthcare system. Firstly, it can prevent the progression to full-fledged T2D, which reduces the burden of this chronic disease for individuals and their families [45]. Secondly, it improves overall health and reduces the risk of diabetes-related complications like heart disease and kidney problems [54]. Moreover, by addressing prediabetes, healthcare resources can be used more effectively, potentially lowering the long-term economic and healthcare costs associated with managing diabetes [55].

## Strength and limitations

This study comes with both strengths and limitations. A significant strength lies in its comprehensive comparison of T2D prevalence and associated risk factors in both urban and rural areas of Bangladesh, using the latest Demographic and Health survey data, which establishes its reliability. Furthermore, we strengthened the study by classifying diabetes and prediabetes based on WHO criteria, while hypertension was defined using the seventh report of the Joint National Committee on Prevention (JNC7). Our ordinal regression model, which has higher statistical power for detecting the effects of explanatory variables compared to conventional logistic regression [56], increases the precision of our findings. However, we must acknowledge certain limitations. One challenge is the presence of a substantial number of missing values and a limited set of risk factors, potentially impacting the comprehensiveness of our results. Additionally, the absence of data on other types of diabetes limits the scope of our study.

## Policy implications

The findings of this study emphasizes critical policy implications for addressing the escalating prevalence of Type 2 Diabetes (T2D) in Bangladesh. To combat urban and rural disparities, targeted educational initiatives are imperative, emphasizing the need for tailored campaigns in urban areas and rural outreach programs addressing specific risk factors. Age-specific interventions are crucial, particularly for the elderly, necessitating tailored healthcare programs and regular health screenings. Addressing socioeconomic disparities is vital, with measures to mitigate economic gaps in urban settings and community-based initiatives in rural areas. Health system preparedness should focus on elderly-focused healthcare and regular screenings. Urban and rural-tailored approaches, including comprehensive health programs in urban regions and promoting healthier work environments in rural areas, are recommended. Acknowledging the substantial prevalence of prediabetes, preventive strategies and health

literacy programs should be prioritized. Data enhancement and continuous monitoring efforts are essential for addressing gaps and ensuring evidence-based policy adjustments. In summary, an integrated, context-specific strategy is essential, necessitating collaboration among policymakers, healthcare providers, and community stakeholders to effectively combat the multifaceted challenges posed by T2D in Bangladesh.

## Conclusion

Our study substantially concludes that individuals who live in metropolitan setting have a higher prevalence of T2D than those from rural areas. Additionally, the higher prevalence of pre-diabetics in both rural and urban areas indicating potential public health burden and rising prevalence of overt diabetes and its related complications. Our findings echo with other studies that the percentage of diabetic patients will increase if an awareness building campaign and some other preventive measures are not taken, considering both rural and urban situations. This study extends our knowledge to compare the prevalence of T2D in both rural and urban settings. Epidemiologists, health policy makers and researchers need to work together to develop a comprehensive program to deal with the looming threat of T2D putting paramount importance on increasing public awareness.

## Author Contributions

**Conceptualization:** Ashis Talukder, Sabiha Shirin Sara.

**Data curation:** Ashis Talukder, Sabiha Shirin Sara, Md. Tanvir Hossain, Chuton Deb Nath, Riaz Rahman.

**Formal analysis:** Ashis Talukder, Sabiha Shirin Sara, Riaz Rahman.

**Methodology:** Ashis Talukder, Sabiha Shirin Sara.

**Resources:** Md. Tanvir Hossain, Sadiq Hussain.

**Software:** Ashis Talukder, Sabiha Shirin Sara, Riaz Rahman.

**Supervision:** Ashis Talukder.

**Validation:** Ashis Talukder, Md. Nazmul Huda.

**Visualization:** Ashis Talukder, Md. Nazmul Huda.

**Writing – original draft:** Ashis Talukder, Sabiha Shirin Sara, Md. Tanvir Hossain, Chuton Deb Nath, Riaz Rahman, Sadiq Hussain, Md. Nazmul Huda.

**Writing – review & editing:** Ashis Talukder, Sabiha Shirin Sara, Md. Tanvir Hossain, Chuton Deb Nath, Sadiq Hussain, Haribondhu Sarma, Md. Nazmul Huda.

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
