## [Decision Letter · Decision Letter 0]

1 Aug 2023

PONE-D-23-16040Rural and Urban Differences in the Prevalence of Type-2 Diabetes and the Associated Risk Factors: Analysis of a Population-Based Nation-wide Cross-sectional Survey in BangladeshPLOS ONE

Dear Dr. Talukder

Thank you for submitting your manuscript to PLOS ONE. After careful consideration, we feel that it has merit but does not fully meet PLOS ONE’s publication criteria as it currently stands. Therefore, we invite you to submit a revised version of the manuscript that addresses the points raised during the review process.

We look forward to receiving your revised manuscript.

Kind regards,

Achyut Raj Pandey, MPH

Academic Editor

PLOS ONE

“NO”

Reviewers' comments:

Reviewer's Responses to Questions

**Comments to the Author**

1. Is the manuscript technically sound, and do the data support the conclusions?

Reviewer #1: Partly

Reviewer #2: Yes

2. Has the statistical analysis been performed appropriately and rigorously? 

Reviewer #1: Yes

Reviewer #2: I Don't Know

3. Have the authors made all data underlying the findings in their manuscript fully available?

Reviewer #1: Yes

Reviewer #2: Yes

4. Is the manuscript presented in an intelligible fashion and written in standard English?

Reviewer #1: No

Reviewer #2: Yes

5. Review Comments to the Author

Reviewer #1: This is a useful work considering the urban-rural variations in prevalence of diabetes. The authors have used data set from demographic health survey and thus has a good sample size with generalizability across the country. However. the manuscript can be improved in a number of ways. Here are some suggestions.

The study uses data from DHS 2017/18. Can you also somehow show the trend in the prevalence and risk factors from previous DHS surveys. There might be some published works. This will provide an insight to the readers about the trend. If the previous DHS surveys do not have blood glucose measurements, you can take reference from previous large sample surveys.

The authors could have done better in describing results and organizing discussion section. It is hard to follow the results in some instances where the reference group is not mentioned.

Interpretation of the findings in the discussion section needs to be improved. You could compare it with more literatures and add implications of the findings. The authors could explore more about characteristics of urban and rural residence to justify the results in the discussion.

The recommendations based on the study findings are too general and vague such as health education programs. The authors could provide specific recommendations linking with the policy and programmatic needs of the country.

The authors have used the term 'insignificant' across multiple instances. It would be better to use 'not significant'

Other comments are attached in the manuscript file. There are queries regarding the classification of physical activity and BMI as well as criteria used to define diabetes status.

Reviewer #2: Rural and Urban Differences in the Prevalence of Type-2 Diabetes and the Associated Risk Factors: Analysis of a Population-Based Nation-wide Cross-sectional Survey in Bangladesh

Thank you for the opportunity to review the paper entitled “Rural and Urban Differences in the Prevalence of Type-2 Diabetes and the Associated Risk Factors: Analysis of a Population-Based Nation-wide Cross-sectional Survey in Bangladesh”. This paper aims to estimate the prevalence of Type-2 Diabetes in rural and urban settings of Bangladesh and identify the risk factors associated with it by locations using the data of Demographic and Health Survey. The authors found that the urban residents had considerable association between diabetes and hypertension.

Here are some of the critical observations while reviewing this paper.

Abstract:

1. Mention DHS after …………. Bangladesh Demographic and Health Survey (DHS)

2. Better to state the 95% CI after point estimates in result section

Introduction:

1. Please check for recent publications for stating the prevalence of Bangladesh. Para 1 and para 2: Please explain staring from global context and then national context. The information about national context popped-up in the middle in para 1.

2. Para 4: Please make sure the citation are in correct order [10, 11, 8] there is [44,42] I somewhere in discussion section as well

Method:

1. Reference might be required in the text related to Sampling design

2. What was the number of total participants in ‘Selection of sample’ ?

3. Bmi/BMI

4. Was there any question related to physical activity in DHS?

5. BMI between 18.5-24.99 is considered as normal in many literatures including WHO recommendation https://www.who.int/europe/news-room/fact-sheets/item/a-healthy-lifestyle---who-recommendations Is there any justification on why the authors have classified BMI between 18.5-22.99 as normal and >= 23 as normal?

6. Is the information about funding source required here?

7. How was the ethical consideration taken while collecting the data?

8. I suggest to present a table here to explain about explanatory variables and how they were categorized throughout to make the readers understand easily.

Results:

1. What the authors mean to say from Primary and secondary education level? Can you explain more about it better in the explanatory variable section?

2. Please make sure you have consistently formatted table regarding the rule of bold texts.

3. I suggest keeping point estimate and 95% CI together XX [ 95% CI: xx-yy] rather than keeping separate columns

Discussion:

1. The authors found that Caffeine use increases the probability of getting T2D . How has the DHS asked about this question? What does caffeine mean here ? is it coffee or tea? Some meta-analysis have shown that caffein intake reduces the incidence of T2D https://link.springer.com/article/10.1007/s00394-013-0603-x If there are options for recording tea or coffee in DHS question, the authors can look if there is significant association between tea and coffee on T2D? Is the tea consumption related to sugar intake?

2. What is the discussion of the authors regarding the those with higher education level were less likely to have diabetes? This contradicts with other previous findings from Bangladesh including https://bmcpublichealth.biomedcentral.com/articles/10.1186/s12889-015-2413-y and others. Authors need to discuss on this finding.

3. There are several studies from Bangladesh and neighbouring countries with similar socio-economic status. It would be better if authors discuss on their findings with the literatures from Bangladesh and its neighbouring countries rather than that from USA

4. For me the last sentence of the Generalizability para came all of sudden. Where is the finding that indicates Urbans are twice more likely to develop T2D than rural population?

5. Please ensure consistency whether you are using T2D or T2DM or Diabetes only throughout the manuscript following the rule of abbreviation

6. PLOS authors have the option to publish the peer review history of their article (what does this mean?). If published, this will include your full peer review and any attached files.

Reviewer #1: No

Reviewer #2: **Yes: **Geha Nath Khanal

NOTE: While revising your submission, please upload your figure files to the Preflight Analysis and Conversion Engine (PACE) digital diagnostic tool, https://pacev2.apexcovantage.com/. PACE helps ensure that figures meet PLOS requirements. To use PACE, you must first register as a user. Registration is free. Then, login and navigate to the UPLOAD tab, where you will find detailed instructions on how to use the tool. If you encounter any issues or have any questions when using PACE, please email PLOS at figures@plos.org. Please note that Supporting Information files do not need this step.

---

## [Author Response · Author response to Decision Letter 0]

12 Sep 2023

Dear Editor (s) and reviewers,

I am delighted to inform you that I, hereby, submit our revised manuscript titled ‘Rural and Urban Differences in the Prevalence of Type-2 Diabetes and the Associated Risk Factors: Analysis of a Population-Based Nation-wide Cross-sectional Survey in Bangladesh’ in your journal for possible publication.

We thank you for giving us the opportunity to revise and resubmit the manuscript. We greatly appreciate the valuable comments from the editor/reviewers and have modified the enclosed manuscript accordingly. In this response letter enclosed, we quote all comments and clarify what we have done in response to each of them. Below, we include reviewers’ comments and our responses to each of the comments.

All changes are highlighted in yellow colour. We hope that our revisions have adequately addressed editor/reviewers’ comments/concerns. We are confident that the revisions have improved the quality of our manuscript. We look forward to hearing from you with a final decision regarding the acceptance of our manuscript.

Regards

Ashis Talukder

The point-by-point response of the reviewer’s comments are mentioned below:

Reviewer #1

This is a useful work considering the urban-rural variations in prevalence of diabetes. The authors have used data set from demographic health survey and thus has a good sample size with generalizability across the country. However. the manuscript can be improved in a number of ways. Here are some suggestions.

1. The study uses data from DHS 2017/18. Can you also somehow show the trend in the prevalence and risk factors from previous DHS surveys. There might be some published works. This will provide an insight to the readers about the trend. If the previous DHS surveys do not have blood glucose measurements, you can take reference from previous large sample surveys.

Response: Thank you for your comments. For better presenting the necessity of this study, the prevalence of diabetes in last decades has been added. Please, look at line number 36-53 in the revised manuscript.

2. The authors could have done better in describing results and organizing discussion section. It is hard to follow the results in some instances where the reference group is not mentioned. Interpretation of the findings in the discussion section needs to be improved. You could compare it with more literatures and add implications of the findings. The authors could explore more about characteristics of urban and rural residence to justify the results in the discussion.

Response: We appreciate your valuable feedback. We have revised both the results and discussion sections based on your comments. Please review the updated result and discussion sections in the revised manuscript. In the discussion section, we have incorporated additional literature for comparison and have highlighted the implications of our findings in greater detail.

3. The recommendations based on the study findings are too general and vague, such as health education programs. The authors could provide specific recommendations linked with the policy and programmatic needs of the country.

Response: Thank you for these valuable comments. We made specific recommendations in our revised manuscript. Please, look at the following paragraph:

“Our findings highlighted a multifaceted approach for controlling the rising prevalence of T2D in both rural and urban areas. This strategy includes promoting healthier eating habits by emphasizing balanced diets and reducing sugar and unhealthy fat consumption, encouraging regular physical activity with accessible facilities, implementing routine health check-ups for early diabetes detection, conducting awareness campaigns tailored to the specific needs of rural and urban populations, improving healthcare accessibility in rural regions, emphasizing blood pressure control and stress management in urban areas, engaging local communities in program design and implementation, training healthcare professionals, advocating for health-promoting policies like taxing sugary drinks, conducting research, and monitoring the effectiveness of interventions. By combining these measures, it is possible to address the diabetes challenge comprehensively while adapting to the unique characteristics of both urban and rural settings.”

4. The authors have used the term 'insignificant' across multiple instances. It would be better to use 'not significant.'

Response: We have taken your comment into account and have consistently used the term 'not significant' throughout the entire manuscript.

5. Other comments are attached in the manuscript file. There are queries regarding the classification of physical activity and BMI as well as criteria used to define diabetes status.

Response: We have addressed all comments mentioned in the manuscript file. We have changed the variable ‘physical activity’ to ‘occupational physical activity status.’ I would like to inform you that there were not any direct questions related to physical activity. We have created the variable from an existing variable “Occupation type”. This process has been followed by several researchers in their previous work. Please, look at the following references:

Talukder, A., Mallick, T.S. On association between diabetes status and stature of individual in Bangladesh: an ordinal regression analysis. Int J Diabetes Dev Ctries 37, 470–477 (2017). https://doi.org/10.1007/s13410-016-0522-5

Dalene KE, Tarp J, Selmer RM, Ariansen IKH, Nystad W, Coenen P, Anderssen SA, Steene-Johannessen J, Ekelund U: Occupational physical activity and longevity in working men and women in Norway: a prospective cohort study. The Lancet Public Health 2021, 6(6):e386-e395.

The full procedure has been described in the “Explanatory variable” part and provides appropriate references. For BMI classification, we followed the Asia-pacific BMI classification guidelines as Bangladesh is an Asian country. The proper reference has been added which supports this classification. Please, look at the following paragraph:

“We created the categorical variable "occupational physical activity status" based on the job description of the respondents [32,33]. If a respondent's occupation involved physical labor or non-sedentary work, we classified them as individuals engaged in physical activity. This category encompasses various occupations, including farmers, fishermen, rickshaw drivers, poultry and livestock raisers, bricklayers, construction workers, road builders, boatmen, and factory workers [32]. Another variable BMI was calculated by the DHS program. They collected data on heights and weights. Heights were measured standing up and weights were measured using SECA scales with a digital display. Then, BMI was calculated as weight in kilograms divided by the square of height in meters. To make our study more meaningful, we then categorized BMI into three categories. In this case, the Asia-Pacific BMI classification had been followed to classify more accurately. BMI with 18.5 kg/m2 was considered underweight, BMI from 18.5 to 22.9 was normal, and BMI ≥ 23 was overweight or obese [34].”

For the classification of diabetes status we were following the guidelines of Americans Diabetes Association (ADA) 

Reviewer #2: Rural and Urban Differences in the Prevalence of Type-2 Diabetes and the Associated Risk Factors: Analysis of a Population-Based Nation-wide Cross-sectional Survey in Bangladesh

Thank you for the opportunity to review the paper entitled “Rural and Urban Differences in the Prevalence of Type-2 Diabetes and the Associated Risk Factors: Analysis of a Population-Based Nation-wide Cross-sectional Survey in Bangladesh”. This paper aims to estimate the prevalence of Type-2 Diabetes in rural and urban settings of Bangladesh and identify the risk factors associated with it by locations using the data of Demographic and Health Survey. The authors found that the urban residents had considerable association between diabetes and hypertension.

Here are some of the critical observations while reviewing this paper.

Abstract:

1. Mention DHS after …………. Bangladesh Demographic and Health Survey (DHS)

Response: It has been updated. Please, see page no. 2, line no. 10 of the revised manuscript.

2. Better to state the 95% CI after point estimates in result section

Response: It has been updated. Please, see page no. 2, line no. 17-18 of the revised manuscript.

Introduction:

1. Please check for recent publications for stating the prevalence of Bangladesh. Para 1 and para 2: Please explain staring from global context and then national context. The information about national context popped-up in the middle in para 1.

Response: Thank you for this valuable comment. We have updated our introduction part based on these comments. Please, look at the introduction part of the revised manuscript.

2. Para 4: Please make sure the citation are in correct order [10, 11, 8] there is [44,42] I somewhere in discussion section as well

Response: The order of citation has been reformed.

Method:

1. Reference might be required in the text related to Sampling design

Response: Reference has been added to the revised manuscript. 

2. What was the number of total participants in ‘Selection of sample’ ?

Response: The number has been specified. Please, look at the following paragraph:

“From a total participant pool of 20,127 individuals, a subgroup of 12,300 met the eligibility criteria for diabetes measurement. These eligible participants were further stratified into two distinct categories: Urban (comprising 4,393 individuals) and Rural (encompassing 7,907 individuals), based on their respective places of residence. This categorization served as the foundation for our subsequent analytical investigations.”

3. Bmi/BMI

Response: It has been updated through whole manuscript.

4. Was there any question related to physical activity in DHS?

Response: We have changed the variable ‘physical activity’ to ‘occupational physical activity status.’ I would like to inform you that there were not any direct questions related to physical activity. We have created the variable from an existing variable “Occupation type”. This process has been followed by several researchers in their previous work. Please, look at the following references:

Talukder, A., Mallick, T.S. On association between diabetes status and stature of individual in Bangladesh: an ordinal regression analysis. Int J Diabetes Dev Ctries 37, 470–477 (2017). https://doi.org/10.1007/s13410-016-0522-5

Dalene KE, Tarp J, Selmer RM, Ariansen IKH, Nystad W, Coenen P, Anderssen SA, Steene-Johannessen J, Ekelund U: Occupational physical activity and longevity in working men and women in Norway: a prospective cohort study. The Lancet Public Health 2021, 6(6):e386-e395.

The full procedure has been described in the “Explanatory variable” part and provides appropriate references. 

5. BMI between 18.5-24.99 is considered as normal in many literatures including WHO recommendation https://www.who.int/europe/news-room/fact-sheets/item/a-healthy-lifestyle---who-recommendations Is there any justification on why the authors have classified BMI between 18.5-22.99 as normal and >= 23 as normal?

Response: For BMI classification, we followed the Asia-pacific BMI classification guidelines as Bangladesh is an Asian country. The proper reference has been added which supports this classification. Please, look at the following paragraph:

“The variable BMI was calculated as weight in kilograms divided by the square of height in meters. To make our study more meaningful, we then categorized BMI into three categories. In this case, the Asia-Pacific BMI classification had been followed to classify more accurately. BMI with 18.5 kg/m2 was considered underweight, BMI from 18.5 to 22.9 was normal, and BMI ≥ 23 was overweight or obese [ref].

[ref] J. U. Lim et al., “Comparison of World Health Organization and Asia-Pacific body mass index classifications in COPD patients,” Int J Chron Obstruct Pulmon Dis, vol. 12, pp. 2465–2475, 2017, doi: 10.2147/COPD.S141295.”

6. Is the information about funding source required here?

Response: We have discarded the information about funding source from method section.

7. How was the ethical consideration taken while collecting the data?

Response: The data used in this study was secondary. The demographic and Health Survey program collected the data and the ethical review committee of DHS maintained the ethical issues. 

8. I suggest to present a table here to explain about explanatory variables and how they were categorized throughout to make the readers understand easily.

Response: In the “Explanatory variable” part, we have described the creation of every independent variable briefly. Moreover, the table 1 shows the categories of the variables with frequency and percentage.

Results:

1. What the authors mean to say from Primary and secondary education level? Can you explain more about it better in the explanatory variable section?

Response: This issue has been updated. Please, look at the following paragraph:

“Within the framework of the Bangladeshi education system, "Primary education" denotes the successful completion of grades 1 to 5, "Secondary education" includes grades 6 to 12, and progression beyond grade 12 is designated as "Higher Education."

2. Please make sure you have consistently formatted table regarding the rule of bold texts.

Response: We have updated all the tables based on the reviewer’s suggestions.

3. I suggest keeping point estimate and 95% CI together XX [ 95% CI: xx-yy] rather than keeping separate columns

Response: We have updated the results section based on the reviewer’s comments. Please, look at the results section of the revised manuscript.

Discussion:

The authors found that Caffeine use increases the probability of getting T2D . How has the DHS asked about this question? What does caffeine mean here ? is it coffee or tea? Some meta-analysis have shown that caffein intake reduces the incidence of T2D

https://link.springer.com/article/10.1007/s00394-013-0603-x If there are options for recording tea or coffee in DHS question, the authors can look if there is significant association between tea and coffee on T2D? Is the tea consumption related to sugar intake?

Response: We appreciate your comment. In response, it's worth noting that the DHS survey asked respondents about their consumption of drinks containing caffeine, such as coffee, tea, cola, or other caffeinated beverages. However, it did not differentiate between tea and coffee. Consequently, we were unable to assess the specific association between tea or coffee consumption and T2D due to the limitations of the survey questions.

2. What is the discussion of the authors regarding the those with higher education level were less likely to have diabetes? This contradicts with other previous findings from Bangladesh including https://bmcpublichealth.biomedcentral.com/articles/10.1186/s12889-015-2413-y and others. Authors need to discuss on this finding.

Response: Thank you for bringing up this concern. We have taken steps to address this issue and have provided a thorough explanation based on our findings. Please review the revised paragraph for further details:

“Our study findings demonstrate an inverse relationship between higher education levels and the incidence of diabetes, both in urban and rural contexts among Bangladeshi individuals. This observation aligns with prior research [38] and implies that education may serve as a protective factor against diabetes by fostering greater health awareness and promoting healthier lifestyle choices [39]. These results underscore the pivotal role of education in the prevention and management of diabetes, emphasizing the importance of educational programs aimed at enhancing health literacy among the Bangladeshi population. Given the apparent significance of higher education in reducing the risk of diabetes, it is essential for policies and interventions to consider educational attainment as a key component in addressing the escalating diabetes burden in Bangladesh and in fostering healthier behaviors across all demographic groups [40]. It is important to note that while education may not directly reduce diabetes, it can substantially improve health literacy and enhance awareness of diabetes-related complications and adherence to dietary recommendations [40]. Further research is warranted to gain deeper knowledge about the mechanisms underlying this relationship and to tailor educational interventions effectively for diverse urban and rural communities.”

3. There are several studies from Bangladesh and neighbouring countries with similar socio-economic status. It would be better if authors discuss on their findings with the literatures from Bangladesh and its neighbouring countries rather than that from USA

Response: It has been updated. Please, look at the discussion section of revised manuscript. 

4. For me the last sentence of the Generalizability para came all of sudden. Where is the finding that indicates Urbans are twice more likely to develop T2D than rural population?

Response: Thank you for your valuable comment and sorry for our mistake. We have updated this paragraph and change the heading ‘Generalizability’ to ‘Policy Implications’.

5. Please ensure consistency whether you are using T2D or T2DM or Diabetes only throughout the manuscript following the rule of abbreviation

Response: We have consistently used T2D throughout the manuscript.

---

## [Decision Letter · Decision Letter 1]

2 Jan 2024

PONE-D-23-16040R1Rural and Urban Differences in the Prevalence and Determinants of Type-2 Diabetes in BangladeshPLOS ONE

Dear Dr. Talukder,

Thank you for submitting your manuscript to PLOS ONE. After careful consideration, we feel that it has merit but does not fully meet PLOS ONE’s publication criteria as it currently stands. Therefore, we invite you to submit a revised version of the manuscript that addresses the points raised during the review process.

We look forward to receiving your revised manuscript.

Kind regards,

Mohammad Nayeem Hasan

Academic Editor

PLOS ONE

Journal Requirements:

Reviewers' comments

**Comments to the Author**

1. If the authors have adequately addressed your comments raised in a previous round of review and you feel that this manuscript is now acceptable for publication, you may indicate that here to bypass the “Comments to the Author” section, enter your conflict of interest statement in the “Confidential to Editor” section, and submit your "Accept" recommendation.

Reviewer #1: All comments have been addressed

Reviewer #3: All comments have been addressed

2. Is the manuscript technically sound, and do the data support the conclusions?

Reviewer #1: Yes

Reviewer #3: Yes

3. Has the statistical analysis been performed appropriately and rigorously? 

Reviewer #1: Yes

Reviewer #3: Yes

4. Have the authors made all data underlying the findings in their manuscript fully available?

Reviewer #1: Yes

Reviewer #3: Yes

5. Is the manuscript presented in an intelligible fashion and written in standard English?

Reviewer #1: Yes

Reviewer #3: Yes

6. Review Comments to the Author

Reviewer #1: The authors have addressed the comments raised during the review process and the paper is now in a good shape. I have now only minor suggestions especially in the discussion section and the table.

Policy implications: redundant information from the opening paragraph of the discussion. Please remove

Policy implications: May be you could start with policy implications which are specific to your findings and not trying to mix-up what the study does not suggest. You could then close the paragraph with a more recommendable statement considering the wider policy environment.

Table: Renaming will be required to describe the caption of the tables. Just mentioning frequency distribution or ordinary logistic regression would not provide enough information to the readers. For instance, table 1 might be renamed as Characteristics of the study participants. Table 3 might be predictors of T2DM across urban and rural areas, the statistical method used could be mentioned below the table. Please go through other literature as a reference.

Reviewer #3: The authors of the manuscript entitled "Rural and Urban Differences in the Prevalence and Determinants of Type-2 Diabetes in Bangladesh"have performed all comments. Thus, the manuscript can be accepted for publication.

7. PLOS authors have the option to publish the peer review history of their article (what does this mean?). If published, this will include your full peer review and any attached files.

Reviewer #1: No

Reviewer #3: No

---

## [Author Response · Author response to Decision Letter 1]

3 Jan 2024

Dear Editor (s) and reviewers,

I am delighted to inform you that I, hereby, submit our revised manuscript titled ‘Rural and Urban Differences in the Prevalence of Type-2 Diabetes and the Associated Risk Factors: Analysis of a Population-Based Nation-wide Cross-sectional Survey in Bangladesh’ in your journal for possible publication.

We thank you for giving us the opportunity to revise and resubmit the manuscript. We greatly appreciate the valuable comments from the editor/reviewers and have modified the enclosed manuscript accordingly. In this response letter enclosed, we quote all comments and clarify what we have done in response to each of them. Below, we include reviewers’ comments and our responses to each of the comments.

All changes are highlighted in yellow colour. We hope that our revisions have adequately addressed editor/reviewers’ comments/concerns. We are confident that the revisions have improved the quality of our manuscript. We look forward to hearing from you with a final decision regarding the acceptance of our manuscript.

Regards

Sabiha Shirin Sara

The point-by-point response of the reviewer’s comments are mentioned below:

Reviewer #1: The authors have addressed the comments raised during the review process and the paper is now in good shape. I now have only minor suggestions, especially in the discussion section and the table.

Policy implications: redundant information from the opening paragraph of the discussion. Please remove.

Policy implications: Maybe you could start with policy implications which are specific to your findings and not trying to mix-up what the study does not suggest. You could then close the paragraph with a more recommendable statement considering the wider policy environment.

Response: Thank you for your suggestions. We have revised the "Policy Implications" section in our manuscript in accordance with the specific findings obtained from our study. Kindly take a moment to review the updated "Policy Implications" section in our revised manuscript.

Table: Renaming will be required to describe the caption of the tables. Just mentioning frequency distribution or ordinary logistic regression would not provide enough information to the readers. For instance, table 1 might be renamed as Characteristics of the study participants. Table 3 might be predictors of T2DM across urban and rural areas, the statistical method used could be mentioned below the table. Please go through other literature as a reference.

Response: Thank you for providing feedback. We have incorporated your suggestions and revised the captions of the tables in our manuscript. Kindly review the updated captions to ensure they align with your recommendations.

Reviewer #3: The authors of the manuscript entitled "Rural and Urban Differences in the Prevalence and Determinants of Type-2 Diabetes in Bangladesh "have performed all comments. Thus, the manuscript can be accepted for publication.

Response: Thank you for your recommendation.

---

## [Editor Report · Decision Letter 2]

18 Jan 2024

Rural and Urban Differences in the Prevalence and Determinants of Type-2 Diabetes in Bangladesh

PONE-D-23-16040R2

Dear Dr. Ashis Talukder,

We’re pleased to inform you that your manuscript has been judged scientifically suitable for publication and will be formally accepted for publication once it meets all outstanding technical requirements.

Kind regards,

Mohammad Nayeem Hasan

Academic Editor

PLOS ONE

---

## [Editor Report · Acceptance letter]

2 Apr 2024

PONE-D-23-16040R2 

PLOS ONE

Dear Dr. Talukder, 

I'm pleased to inform you that your manuscript has been deemed suitable for publication in PLOS ONE. Congratulations! Your manuscript is now being handed over to our production team.

Kind regards, 

on behalf of

Dr. Mohammad Nayeem Hasan 

Academic Editor

PLOS ONE